# A New Look at the Structure and Thermal Behavior of Polyvinylidene Fluoride–Camphor Mixtures

**DOI:** 10.3390/polym14235214

**Published:** 2022-11-30

**Authors:** Konstantin V. Pochivalov, Andrey V. Basko, Tatyana N. Lebedeva, Anna N. Ilyasova, Georgiy A. Shandryuk, Vyacheslav V. Snegirev, Vladimir V. Artemov, Alexander A. Ezhov, Yaroslav V. Kudryavtsev

**Affiliations:** 1Krestov Institute of Solution Chemistry, Russian Academy of Sciences, Akademicheskaya ul. 1, Ivanovo 153045, Russia; 2Institute of Macromolecular Compounds, Russian Academy of Sciences, Bolshoy pr. 31, St. Petersburg 199004, Russia; 3Topchiev Institute of Petrochemical Synthesis, Russian Academy of Sciences, Leninskii pr. 29, Moscow 119991, Russia; 4Faculty of Physics, Lomonosov Moscow State University, Leninskie Gory 1–2, Moscow 119991, Russia; 5Shubnikov Institute of Crystallography, Federal Scientific Research Center “Crystallography and Photonics”, Russian Academy of Sciences, Leninskii pr. 59, Moscow 119333, Russia; 6Frumkin Institute of Physical Chemistry and Electrochemistry, Russian Academy of Sciences, Leninskii pr. 31, Moscow 119071, Russia

**Keywords:** semicrystalline polymer, camphor, phase diagram, thermally induced phase separation, morphology, polymorphous transition

## Abstract

An experimental quasi-equilibrium phase diagram of the polyvinylidene fluoride (PVDF)–camphor mixture is constructed using an original optical method. For the first time, it contains a boundary curve that describes the dependence of camphor solubility in the amorphous regions of PVDF on temperature. It is argued that this diagram cannot be considered a full analogue of the eutectic phase diagrams of two low-molar-mass crystalline substances. The phase diagram is used to interpret the polarized light hot-stage microscopy data on cooling the above mixtures from a homogeneous state to room temperature and scanning electron microscopy data on the morphology of capillary-porous bodies formed upon camphor removal. Based on our calorimetry and X-ray studies, we put in doubt the possibility of incongruent crystalline complex formation between PVDF and camphor previously suggested by Dasgupta et al. (Macromolecules 2005, 38, 5602–5608). We also describe and discuss the high-temperature crystalline structure of racemic camphor, which is not available in the modern literature.

## 1. Introduction

At present, there is a growing interest in the preparation of polymeric capillary-porous materials, such as membranes for liquid filtration [1,2,3,4,5,6,7,8] and gas separation [9,10,11], porous pellets for immobilization of pharmaceuticals [12,13,14] or nanoparticles [15,16], scaffolds for tissue engineering [17,18], etc. [19,20,21,22]. From the variety of polymers used for the production of such materials, considerable attention is attracted by polyvinylidene fluoride (PVDF) [1,2,3,15,19,20] due to its low cost, good chemical and thermal resistance, and decent mechanical properties [1]. Capillary-porous bodies were obtained from the blends of PDVF with thermodynamically good [23,24,25,26,27] and poor [28,29,30,31] liquid solvents. At the same time, studies focused on the preparation of such materials from the mixtures of this polymer with crystalline substances are still scarce [32,33,34,35].

Dasgupta et al. [33,34,35] prepared multiporous PVDF materials with micro-, meso-, and macropores using camphor as a crystalline diluent, which could be easily removed from the samples by sublimation, thus bypassing the extraction step. As an interesting feature, they reported on the formation of crystalline complexes (compounds) between PVDF and camphor and constructed the corresponding phase diagram (Figure 1) based on the differential scanning calorimetry (DSC) data and conclusions made from synchrotron X-ray experiments, optical observations, and molecular modelling for this system.

Our analysis of these publications raised some serious questions. Firstly, the diagram in Figure 1 contains several curves reflecting hypothetical processes (melting of camphor excess, melting of crystalline complexes with different structure, etc. [33,34]), the thermal effects of which have not been recorded. At the same time, it includes the binodal of liquid–liquid equilibrium above the camphor melting temperature, which should be characterized by a much lower thermal effect than melting. Moreover, this binodal is present for rapid heating (40 °C/min) and absent at a much slower (2 °C/min) heating rate. 

Secondly, the formation of solid solutions of PVDF in camphor with the polymer concentration increasing with temperature, as well as the formation of crystalline complexes between camphor and PVDF, seems doubtful in the absence of strong intermolecular interactions. The cocrystallization of PVDF with camphor should change the crystal structure of the polymer, and the peaks corresponding to the α-phase of pure PVDF are hardly expectable at high concentrations of camphor. Nevertheless, they were observed in ref. [33,34] for the mixtures containing up to 80 wt% of camphor. It should also be noted that in two consequent papers, different descriptions of the complex stoichiometry are provided. In ref. [33], the complexes are thought to include four monomer units of PVDF and five camphor molecules, whereas in ref. [34], two different complexes are described, which contain two and four PVDF units per camphor molecule, respectively. 

Thirdly, nitrogen adsorption/desorption porosimetry seems to considerably overestimate the volume of pores smaller than 20 nm in a polymer matrix. We observed this effect for the capillary-porous bodies of low-density polyethylene [36] and isotactic polypropylene [37] prepared via thermally induced phase separation (TIPS) from solutions in dimethyl terephthalate and camphor, respectively. In both cases, a rather high specific volume of nanopores (up to 0.3 cm^3^/g), which is comparable with the overall porosity of the samples, was obtained. We have assumed that this value corresponds to nitrogen adsorbed on all nanoscale asperities on both internal and external surfaces of the samples. Therefore, it should not be used for assessment of the quantity and size of nanoscale pores in the capillary-porous sample formed via TIPS. 

All the above factors cast into doubt the conclusions made by Dasgupta et al. and motivated us to carry out our own structural and thermal studies of the PVDF–camphor mixtures. An interpretation of the data obtained for this particular system can be made in the framework of our concept proposed for the mixtures of semicrystalline polymers with low-molar-mass (LMM) compounds [37,38,39,40]. This concept suggests that the phase diagrams of such systems should be supplemented with an extra boundary curve reflecting the dependence of LMM substance solubility in the amorphous polymer regions on temperature. This boundary curve was constructed for a number of polyolefin (low- and high-density polyethylene and polypropylene) mixtures with n-alkanes [40], alkylbenzenes [41], dialkyl phthalates [39], 1,2,4,5-tetrachlorobenzene [42], thymol [43], camphor [37], and dimethyl terephthalate [36] and for poly-3.3-bis(azidomethyl)oxetane with energetic plasticizer [44] using a quasi-equilibrium optical method. The corresponding phase diagrams allowed us to discuss the thermal behavior and structural evolution of the above mixtures in a thermodynamically consistent manner.

While preparing this paper, we found that the crystalline structure of camphor at high temperatures, between ca. 100 °C and its melting point at 170 °C, is missing in the major repositories (Cambridge Crystallographic Data Centre (CDCC) and Crystallography Open Database). Several papers report that at room temperature, racemic camphor forms the hexagonal (hex) crystalline phase with known lattice parameters [45,46,47,48,49,50] and that upon heating in the interval from 75 °C to 100 °C, it undergoes a transition to the face-centered cubic (fcc) phase with a single lattice parameter of 10.1 Å [34,45,47,48,49,51,52,53]. However, none of these papers contain the original X-ray diffraction (XRD) data, which inspired us to conduct our own study of the hex and fcc crystalline phases of camphor.

The rest of the paper is organized as follows. After the Materials and Methods section, we use our quasi-equilibrium optical method to construct a reliable phase diagram of PVDF–camphor. The states of the mixtures at different compositions and temperatures are discussed in detail. Then, we analyze the DSC curves and interpret peaks recorded at heating and cooling experiments. The DSC data are used to obtain a dynamic phase diagram of the system, which is then compared with the diagram obtained by the optical method. After that, we present the results of a new study of the crystalline structure of camphor and try to interpret the XRD data by Dasgupta et al. [33,34,35] in a consistent manner. The phase diagram is also used to explain the structural evolution of the PVDF–camphor mixtures during cooling from a homogeneous state to room temperature. Finally, we analyze the internal morphology of the capillary-porous PVDF bodies obtained after camphor removal. The paper is concluded with a summary of our findings.

## 2. Materials and Methods

PVDF (Solef 6020, Solvay Specialty Polymers, *M*_w_ ≈ 650 kg/mol according to ref. [54,55,56]) with a melt flow index of 4.5 g/10 min measured at 230 °C under a load of 21.6 kg (DIN EN ISO 1133:2005), crystallinity degree of 73.5% found by DSC at first heating using 104.7 J/g [57] as the melting enthalpy of PVDF at 100% crystallinity, density of 1.78 ± 0.01 g/cm^3^ found picnometrically at 25 °C, and full amorphization temperature of 175.8 ± 0.3 °C (end of the DSC melting peak) and camphor (racemic, 96%, Acros Organics, Geel, Belgium) with a melting temperature of 170 ± 2 °C (onset of the DSC melting peak) and boiling point of 204 °C were used without additional purification.

Experimental phase diagrams were constructed using the optical and DSC methods. The optical technique [58] consists of observation, through a horizontal microscope, of changes in the state of a binary system during its stepwise heating in a sealed ampoule. This method makes use of a difference in the refractive indexes of coexisting phases. In our case, it allows one to determine, with an accuracy of ±0.5 °C, the temperatures at which the state of an initially two-phase system containing a PVDF pellet and camphor crystals is qualitatively changed under heating from room temperature at a very slow rate of ca. 1 °C per day.

DSC thermograms for PVDF, camphor, and their mixtures of different compositions were recorded on a DSC823e/400 calorimeter (Mettler-Toledo, Greifensee, Switzerland) at a scanning rate of 10 °C/min under argon flow and sample mass of 3–7 mg. Calibration standards were *n*-hexane (temperature and enthalpy of melting of −94 °C and 151.8 J/g, respectively), distilled water (0 °C, 334.5 J/g), and indium (156.6 °C, 28.45 J/g). The measurement accuracy was within ±0.3 °C for temperature and ±1 J/g for enthalpy. Sealed crucibles containing PVDF/camphor compositions were cooled down from room temperature to 0 °C, heated up to 210 °C, kept at that temperature for 30 min, then again cooled to 0 °C and heated up to 210 °C for the second time. The first heating and annealing at 210 °C provided homogenization of the samples, while cooling to room temperature led to an increase in the contact area between the components, thus accelerating diffusion in the course of further heating. At both the first and second heating stages, the DSC curves were recorded. Some samples were treated at a slower scanning rate of 2 °C/min, and some were cooled down to –90 °C (instead of 0 °C) between the heating cycles.

Optical investigations of the structure of PVDF–camphor mixtures were performed on an RNMK-05 microscope (VEB MLW Analytik, Dresden, Germany, 100x zoom) equipped with a compact Boetius heating stage (heating/cooling rate of 5 °C/min). A small amount of the mixture was placed into the glass cell and covered with another glass in order to form a film of uniform thickness upon melting. The cell was covered by an extra glass to prevent camphor losses via sublimation or evaporation. Some of the mixtures were also studied under a Micromed C-11 microscope (Micromed, Moscow, Russia, 800x zoom) after cooling to room temperature.

Capillary-porous bodies were prepared as follows. A PVDF–camphor mixture homogenized at 210 °C was poured onto a glass plate heated up to the same temperature, covered with another hot glass, and cooled down to room temperature in air. Then, camphor was extracted into an isopropanol bath (1:50 by mass, 5 h), and the samples were dried in air for 48 h at room temperature.

For morphological studies, the sample was fixed on a scanning electron microscopy (SEM) holder and cooled down to the temperature of liquid nitrogen. Then, it was cleaved perpendicular to the surface of the film with a knife and placed in a JSM-7401F SEM device (JEOL, Tokyo, Japan) equipped with a cold cathode field emission gun. Secondary electron images were acquired using the lower detector at an accelerated voltage of 1 kV and working distance of 8–9 mm without conductive coating. Images were taken from the surface areas without defects that could arise from sample fixing and cleavage.

An X-ray study of the crystal lattice of camphor at different temperatures was performed using a STADI P powder diffractometer (STOE, Darmstadt, Germany) with a capillary sample holder and Co Kα1 radiation. Both the IP-PSD imaging plate detector for preliminary measurements at room temperature and the scintillation counter point detector for fine measurements were used. Borosilicate glass capillary tubes (Capillary Tube Supplies, Withiel, UK) with a 1.0 mm outside diameter plugged with polytetrafluoroethylene sealing tape to prevent camphor losses were used as sample cases. The measurements at elevated temperatures were carried out using a hot air rework station Element 858 and an independent temperature measuring device consisting of a Type K (chromel–alumel) thermocouple connected to an Mastech Digital M838 multimeter (MGL, Taipei, Taiwan). Preliminary measurements demonstrated that the accuracy of temperature stabilization was ±2.5 °C.

Modelling of XRD powder patterns for the fcc crystalline phase of racemic (D,L) camphor was performed as follows. Atom coordinates in the molecules of D-camphor and L-camphor were found using ChemSketch 2019.1.3 freeware (ACD/Labs) after 3D optimization. Phase plasticity was introduced using LibreOffice freeware (TDF). To that end, the centers of mass of both enantiomers were found and placed at the origin of the reference system. Then, the molecules were successively rotated by three Euler angles, the values of which were chosen at random, and placed at 14 nodes of the fcc crystal lattice with a period of 10.1 Å so that each node was occupied simultaneously with two different enantiomers and so that all molecules of the same chirality were oriented in the same way. This procedure of specifying the molecular coordinates was repeated to create twenty independent crystallographic information files (CIFs), which were used for the powder pattern calculation with Mercury 4.1.0 freeware (CDCC). The files are presented in Appendix A. Calculated powder patterns were averaged using LibreOffice and recalculated to the XRD intensities for various interplanar distances *d_hkl_*. Modelling of XRD powder patterns for the hex crystalline phase of camphor was performed using Mercury 4.1.0 freeware with the CIF file for L-camphor prepared by ab initio molecular dynamics in ref. [50]. Simulated XRD patterns were used for peak identification in the experimental XRD patterns.

## 3. Results and Discussion

### 3.1. Phase Diagram by the Optical Method

The experimental phase diagram of PVDF–camphor constructed by the optical method is presented in Figure 2 and illustrated by optical photographs 1–6 at points K, K_1_, K_2_, K’, K_1_’, and K_2_’. It is topologically similar to the diagrams of isotactic polypropylene with camphor [37] and of low-density polyethylene with 1,2,4,5-tetrachlorobenzene [59] studied by us previously but significantly differs from the diagram shown in Figure 1 and from some other phase diagrams of the mixtures of semicrystalline polymers with LMM crystalline substances reported in the literature [60,61,62,63,64,65,66,67,68]. Thus, it is worth discussing it in more detail.

The optical method enables one to detect qualitative changes in the system state in the course of its stepwise heating. Let it initially consist of a PVDF pellet and camphor crystals at room temperature (photo 1 and photo 4). 

For a polymer-rich mixture containing 80 wt% of PVDF, heating from point K results in a gradual increase in the pellet volume and a decrease in the amount of crystalline camphor. As was shown in ref. [69], both vapor-phase and direct mechanisms contribute to the process of camphor dissolution in the amorphous regions of PVDF. When the temperature corresponding to point K_1_ is attained, no more camphor crystals are visible, whereas the polymer remains turbid due to its crystallinity (photo 2). Further heating diminishes the turbidity, and at point K_2_, the system becomes fully transparent (photo 3). However, the pellet still keeps its shape due to the very high viscosity.

In a polymer-lean mixture with 20 wt% of PVDF, heating from point K’ (photo 4) also results in a swelling of the polymer pellet and a decrease in its turbidity. At point K_1_’, the pellet becomes liquid so that the remaining camphor crystals coexist with a camphor solution in the polymer melt (photo 5). Further heating gradually lowers the amount of camphor crystals, and finally, the system transforms into a single-phase solution of camphor in the polymer melt (photo 6). 

The boundary curves BD, BC, AB, and EB in Figure 2 are plotted according to the temperatures, at which the events in photos 2, 3, 5, and 6 are detected for the mixtures of different compositions. The ABC curve describes the dependence of the full polymer amorphization temperature on the composition of the initial mixture. The EBD curve reflects the temperature dependence of camphor solubility in the amorphous regions of semicrystalline PVDF (along the BD segment) or in the polymer melt (along the EB segment). These boundary curves delineate four domains in the phase diagram: (I) single-phase homogeneous mixtures of PVDF melt and camphor; (II) single-phase solutions of camphor in semicrystalline PDVF; (III) two-phase systems composed of camphor crystals and the camphor solution in semicrystalline PVDF; (IV) two-phase systems containing camphor crystals and the camphor solution in melted PVDF. Note that the systems in domain II are heterogeneous at the nano level, being composed of polymer crystallites and swollen amorphous regions. They are similar to physical gels, in which crystallites play the role of crosslinks interconnected by tie chains. The mass fraction of crystallites in a mixture of a given composition decreases on heating and reaches zero at the BC curve.

Since semicrystalline polymers are generally considered non-equilibrium or metastable systems [70], solutions of LMM substances in such polymers should be referred to as non-equilibrium as well. Therefore, the term “phase” is, strictly speaking, inappropriate for the systems below the ABC curve. That is why it would be more correct to characterize the states in domains II and III as “macroscopically homogeneous” and “macroscopically inhomogeneous” instead of “single-phase” and “two-phase”. The absence of a true equilibrium state in semicrystalline polymers hinders the application of higher characterization methods reported for the complex organic [71] and inorganic [72,73] systems exhibiting crystallinity.

It also calls into question the applicability of the Gibbs–Thomson equation [74], which relates broad melting peaks of polymers to the thickness distribution of their crystalline lamellas. In the last few years, more and more evidence [75,76,77,78,79,80,81,82] contributing to the opinion that the distribution of lamellar thickness cannot fully explain the melting behavior of semicrystalline polymers appeared. For example, crystallites of the same size can melt at different temperatures depending on the thickness of adjacent amorphous regions [75]. Polymer swelling affects the destruction of crystallites during uniaxial film stretching [77]. An increase in chain mobility with temperature can lead to the reorganization of basal surfaces of lamellas [82]. These data are well in line with our idea [38,83] about the thermomechanical equilibrium attained between the mechanical strength of the most defective crystallites and internal stress in the amorphous regions of semicrystalline polymers. Since the state of the amorphous regions is affected by both temperature and the swelling degree of the polymer, the same factors should be responsible for the destruction of crystallites. In our stepwise heating experiments, this process manifests itself through a decrease in the sample turbidity. In domain III, this is caused by both an increase in temperature and an increase in swelling degree, and in domain II, where the polymer swelling degree is constant, amorphization is due to an increase in temperature only.

Thus, the topology of the phase diagram in Figure 2 and of similar diagrams presented in ref. [37,42,59,69] differs from that of classical eutectic phase diagrams [61,62,63,64,65,66,67,68], since we do not discard the swelling ability of semicrystalline polymers. Regarding Figure 1, we can conclude that no liquid–liquid equilibrium exists in this system above the camphor melting temperature. Since a very slow heating rate of ca. 1 °C per day was used, we have to suppose that the interpretation of high-temperature endothermic peaks at DSC curves as the thermal effects of mixing melted PVDF with melted camphor [33,34] is incorrect. Also note that in the above papers, no effect of camphor melting was recorded, which should be much higher than the thermal effect of two liquids mixing.

In order to clarify whether the crystalline complexes between PVDF and camphor can be formed, we performed our own DSC study of the corresponding mixtures.

### 3.2. Analysis of the DSC Data

The results of the first heating experiments are presented in Figure 3. The endotherm for pure PVDF (curve 1) has two peaks, which is probably not related to the crystallite thickening since the heating rate was 10 °C/min. It can correspond to two groups of crystallites that differ in the degree of their perfection and thus melt at different temperatures. Such peaks are also observed for the mixtures of PVDF with camphor (curves 2–10), and their areas and maximum temperatures decrease with an increase in the camphor concentration in the initial blend from 10 wt% to 90 wt%.

For pure camphor (curve 11) and the mixtures containing no more than 70 wt% of PVDF (curves 4–10), a high-temperature endotherm is observed. We associate this peak with two simultaneous processes: dissolution of camphor in the PVDF melt already containing some dissolved camphor and melting of camphor that does not dissolve in the polymer due to kinetic limitations. Since the thermal effect of melting dominates, the DSC experiments report that the position of this peak is nearly constant. In the absence of kinetic limitations in much slower optical experiments, a non-horizontal dissolution EB curve is recorded (Figure 2). The same camphor-rich systems (curves 4–11) reveal another small endothermic peak at 90 ± 2 °C. As follows from the literature [48,53,84], this peak reflects a thermal effect of the high-temperature solid–solid (hex→fcc) transition in camphor. We will consider this transition in more detail when discussing the XRD results.

The results of the second heating experiments are presented in Figure 4. After erasing the thermal history, PVDF reveals a single broad endothermic peak. This peak, marked with blue, is observed for all PVDF–camphor mixtures. It combines the thermal effects of PVDF thermomechanical amorphization and the dissolution of camphor in its amorphous regions. Its maximum decreases from 170 °C to 133 °C with an increase in the camphor concentration until it reaches 50 wt% (curves 1–6) and then remains nearly constant (half green/half blue points). Curves 3–5 also contain a nearby peak, marked with green, the temperature of which slightly grows from 127 °C to 133 °C with the camphor concentration. Based on our previous study of isotactic polypropylene–camphor mixtures [37], we can identify the green peak with the melting of nanoscale camphor crystals, which are formed under confinement of a semicrystalline polymer matrix. The melting temperature of these crystals is about 130 °C irrespective of the polymer type, which corresponds to a size of ~10 nm according to the Gibbs–Thomson equation [85]. Note that an endotherm with an approximately constant peak position at 130 °C was detected in ref. [33,34] and interpreted as a melting curve for one of the types of PVDF–camphor complexes discussed in these papers.

At high (≥50 wt%) concentrations of camphor, green and blue peaks merge into one peak. However, we can again record two separate peaks even at 70 wt% of camphor if the scanning rate is dropped from 10 °C/min to 2 °C/min (Figure 4c). The position of the green peak that reflects the melting of small camphor crystals remains unchanged, while the blue peak that describes PVDF amorphization is expectably shifted to higher temperatures [86,87].

It is also instructive to follow changes in the intensity of the blue peak. A decrease in the polymer concentration (curves 1–5) makes it smaller because the heat flux recorded by DSC is normalized per total mass of the mixture. When the blue peak merges with the green one, the intensity of the combined peak becomes markedly higher. However, a further decrease in the polymer concentration results both in the reduction in the contribution of PVDF amorphization and in the decrease in the amount of camphor nanocrystals, which can be formed inside the polymeric matrix only. This naturally leads to a decrease in the half green/half blue peak intensity for curves 7–10.

Figure 4 also contains red and light blue peaks related to camphor. The red peak corresponds to the EB curve in the optical phase diagram (Figure 2) and reflects the dissolution of camphor crystals in the PVDF melt. Both the intensity and temperature of this peak expectably decrease with an increase in the polymer concentration. The light blue peak, which is observed at the same temperature for the first and second heatings, describes the solid–solid (hex→fcc) transition in camphor. As follows from the literature [48,53,84], there should be one more polymorphic transition of orthorhombic (orth)→hex type, which was indeed detected by us between −60 °C and −70 °C in additional experiments when the mixtures were cooled down to −90 °C between the first and second heatings (Figure 4b).

The results of DSC cooling experiments are presented in Figure 5. The mixtures containing from 10 wt% to 50 wt% of the polymer (curves 6–10) reveal two exothermic peaks. The high-temperature peak reflects camphor crystallization from its liquid mixture with PVDF. Its intensity and temperature both increase with the increase in the camphor concentration. The low-temperature peak describes two processes, i.e., PVDF crystallization from its liquid mixture with camphor and the formation of small camphor crystals in the amorphous polymer regions. Its intensity increases with the increase in the polymer concentration, whereas the onset temperature is independent of the mixture composition. For the mixtures containing more than 50 wt% of the polymer (curves 2–5), only the latter peak is recorded. The onset temperature of this peak decreases from 127 °C to 97 °C with the increase in camphor concentration in the initial mixture.

### 3.3. Comparison of the Optical and DSC Data

DSC data discussed in the previous section were used to construct the dynamic phase diagram plotted in Figure 6. The E_1_B_1_, A_1_B_1_, B_1_C_1_, B_1_H, and FG curves are based on the maximum temperatures of the endothermic peaks at the second heating and the E_2_B_2_ and A_2_B_2_C_2_ curves on the exotherm onset temperatures at cooling. They describe compositional dependences of the following temperatures: (E_1_B_1_) dissolution of camphor in melted PVDF, (B_1_C_1_) full polymer amorphization, (B_1_H) melting of small camphor crystals, (A_1_B_1_) simultaneous full polymer amorphization and melting of small camphor crystals, (FG) polymorphic hex→fcc transition in pure camphor, (E_2_B_2_) camphor crystallization out of the liquid mixture, and (A_2_B_2_C_2_) PVDF crystallization.

By comparing the positions of the corresponding boundary lines obtained by (color curves) calorimetric and (black curves) optical methods, we conclude that DSC in the heating regime systematically underestimates the temperatures of full polymer amorphization (ABC curve) and slightly overestimates the temperatures of camphor dissolution (EB curve). This is a typical feature for such kinds of binary systems [38], which mainly stems from the dynamic nature of the DSC method. The boundary lines from heating experiments are considerably higher on the temperature axis than the corresponding curves from cooling runs, which is due to the kinetic factors in polymer crystallization. The BD solubility curve cannot be constructed by DSC at all, since this method does not allow one to distinguish the simultaneous processes of camphor dissolution in the amorphous regions of PVDF and polymer amorphization. At the same time, DSC enriches the phase diagram with the A_1_B_1_H curve that describes the melting of nanoscale camphor crystals formed within the polymer matrix. Its A_1_B_1_ segment coincides with the PVDF full amorphization curve, whereas the B_1_H segment is a separate curve. Note that its presence does not affect the conclusions from the optical diagram because the system in domain III (Figure 2), both above and below the B_1_H curve, contains camphor crystals and camphor solution in the amorphous regions of PVDF.

The authors of ref. [33,34] qualitatively obtained the same DSC curves as in this study but presented an alternative interpretation that led to the intricate phase diagram in Figure 1. They also performed synchrotron X-ray experiments on the PVDF–camphor mixtures. Without the intention to repeat their experiments completely, we will nonetheless show in the next section that an interpretation of the XRD experiments is also possible without assuming the formation of PVDF/camphor crystalline complexes.

### 3.4. Solid–Solid Phase Transition in Camphor

Speculations on the formation of crystalline complexes of PVDF with camphor in ref. [33,34,35] are based on the observation that the diffractograms obtained at elevated temperatures contain reflections that are absent on the diffractograms of pure components at room temperature. However, an alternative explanation is possible if one takes into account that solid camphor undergoes a polymorphic transition (hex→fcc) in the temperature range from 75 to 100 °C [34,45,47,48,49,51,52,53]. So far, we have carried out XRD experiments at room temperature and at 120 °C, and in parallel, simulated XRD on the molecular models of the corresponding crystalline phases of camphor. Both phases belong to the family of orientationally disordered (plastic) crystals [88,89], where the molecules can freely rotate relative to the crystal lattice nodes. The procedure for preparing the model structures is described in the Materials and Methods section, and the result of its implementation is illustrated in Figure 7. The superpositions of four out of the twenty crystal cells (see CIFs in Appendix A) used to calculate a pattern of the high-temperature fcc phase of camphor are shown in Figure 7a. For comparison, Figure 7b contains a crystal cell of the room-temperature hex phase of camphor taken from ref. [50], in which the influence of rotational freedom of camphor molecules on the XRD pattern was considered in detail. For better perception, Figure 7 also shows the positions of the centers of mass of camphor molecules in the crystalline cells of the respective phases, which demonstrate a significant difference between the crystalline structures.

The experimental XRD patterns are depicted in Figure 8a and compared with the simulation data in Figure 8b. Note that the results obtained by different methods were presented in different coordinates (intensity vs. the double glancing angle 2θ for ordinary XRD and log(intensity) vs. the transfer momentum *q* for synchrotron radiation XRD). 

Because of that, we have recalculated the results to find the interplanar spacing *d_hkl_* characteristic of the crystal structure, which is independent of wavelength and the XRD method. One can see from Figure 8 that an increase in the temperature from 25 °C to 120 °C leads to the disappearance of 222, 201, 221, 002, 200, and 220 reflections from camphor with the hex crystalline lattice and to the appearance of 200, 002, and 111 reflections from camphor with the fcc crystalline lattice. A comparison of the experimental data with the simulation results shows good qualitative agreement. Some deviations in the absolute peak positions can be caused by a slight displacement of the sample relative to the axis of the goniometer. Nevertheless, the distances between different peaks describing the fcc and hex phases of camphor are the same in the experiment and in simulations.

In Figure 9, our data are compared with the results of Dasgupta [33,34] recalculated to the same dependences of the XRD intensity on the interplanar distance *d_hkl_*. We can conclude that the appearance of at least one, the most prominent, peak corresponding to *d_hkl_* = 5.6–5.8 Å in the presence of PVDF can be caused by the hex→fcc phase transition in camphor upon its heating in the synchrotron experiments and/or by a change in the predominant orientation of hex camphor phase crystals at room temperature. In both cases, there is no need to assume that the cocrystallization of PVDF with camphor takes place. Moreover, were such crystal complexes real, it would be strange to observe the peaks corresponding to the crystalline α-phase of PVDF in the *d_hkl_* = 4.3–5.2 Å range up to 80 wt% of camphor (Figure 9).

### 3.5. Crystallization in the Pre-Homogenized Mixtures

Homogeneous mixtures of PVDF and camphor were prepared by heating the sealed ampoules with the components up to 210 °C. Their evolution at subsequent cooling to room temperature can be predicted using the phase diagram in Figure 2. If a mixture with the polymer mass fraction *w*_2_ < *w*_2B_ = 0.5 is taken, then crossing the EB curve would launch the crystallization of camphor from its liquid mixture with PVDF. Further cooling into domain IV should result in the formation of a dispersion of camphor crystals in a liquid mixture of camphor and PVDF. When it is cooled down to the AB line, the polymeric component would start to crystallize. In that state, its volume fraction in the liquid mixture always equals *w*_2B_, irrespective of the initial mixture composition, provided *w*_2_ < *w*_2B_. The system transforms into a solution of camphor in the amorphous regions of PVDF, which behaves as a physical gel with crystallites as crosslinks of the 3D network. Since the growth of camphor crystals is now hindered by the solid-like polymer matrix, the subsequent cooling into domain III results in the formation of smaller crystals than in domain IV.

Now consider the experimental data obtained with a hot-stage optical microscope for the mixture containing 35 wt% of PVDF, which is cooled down at an average rate of 5 °C/min. Figure 10a shows that at 180 °C, the system is still homogeneous. At 160 °C, fern-shaped camphor crystals become visible (Figure 10b), and they continue to grow at lower temperatures (Figure 10c,d). At 108 °C, the surrounding solution becomes turbid (Figure 10e) due to polymer crystallization. Subsequent cooling enhances light scattering, and hence, even well-visible contours of the camphor crystals become blurred (Figure 10f), which can be related to the formation of small camphor crystals inside the polymer matrix.

The cooling of homogeneous PVDF–camphor mixtures with the composition *w*_2_ > *w*_2B_ should first lead to polymer crystallization as soon as the BC curve is reached. In that case, a physical gel is formed throughout entire volume of the sample. Subsequent cooling into domain II would increase polymer crystallinity, and crossing the BD curve would cause microphase separation of the gel and camphor crystallization in domain III.

The study of the mixture containing 75 wt% of PVDF, which is homogeneous at 180 °C (Figure 11a), reveals nucleation (Figure 11b) and growth (Figure 11c–f) stages of polymer crystallization. The spherulites grow until they start to impinge on each other (Figure 11e,f). At 114 °C, a macroscopically uniform gel is formed throughout the entire sample. This gel is expected to contain 25 wt% of camphor that remains dissolved in the amorphous regions of the polymer. At 106 °C, the microphase separation starts, and camphor forms small crystals within the polymer matrix (Figure 11g). The amount of such crystals increases during subsequent cooling (Figure 11h).

We can conclude that both polymer-lean and polymer-rich mixtures follow the phase diagram predictions, though the crystallization of PVDF is detected at temperatures that are more than 30 °C lower than the temperature of full polymer amorphization in these systems. This is partly due to the fact that it takes some time for the polymer crystallites to grow until they are visible in the optical microscope, but the main reason is a low crystallization rate, characteristic of polymers in general. Camphor first crystallizes (Figure 10b) at 160 °C, which is very close to the point on the EB curve in Figure 2 that corresponds to a mixture containing 35 wt% of PVDF.

### 3.6. Morphology of the Capillary-Porous Bodies

We used scanning electron microscopy to study the internal structure of samples upon cooling from homogeneous mixtures at 210 °C to room temperature and removal of camphor. The remaining pores reproduce the shape of the camphor crystals that existed in their place.

Figure 12 reveals that a capillary-porous body prepared from the mixture with 28 wt% of PVDF contains two types of porous structures. First, there are large pores (Figure 12a,b) formed by fern-shaped camphor crystals, which grow in domain IV of the phase diagram in Figure 2, when the polymer is fully melted. Their dimensions in the longitudinal and transverse directions can exceed 100 and 10 μm, respectively. The pores of the second type are much smaller (200 nm and less), being visible on the external side (Figure 12c) and the cross-section (Figure 12d) of the polymer matrix. They are formed within domain III of the phase diagram.

An increase in the polymer mass fraction in the initial mixture from 28 wt% to 45 wt% does not qualitatively change the morphology (Figure 13) because both these systems are situated to the left of point B in the phase diagram. However, the size of the large pores decreases, since domain IV becomes narrower as point B is approached, and camphor crystals have less time to grow without restrictions imposed by polymer crystallites (Figure 13a). The morphology of the cross-section of a wall between large pores is visible at the top right of Figure 13b. One can see that small pores forming cellular structures are uniformly distributed inside the wall. They are formed in the course of microphase separation in the physical gel formed by a semicrystalline polymer. Camphor crystals in these pores that reach 1 μm in size cause blurriness of the optical image shown in Figure 10f. The smallest pores, with an average size of about 10 nm according to DSC, are located between polymer lamellas, i.e., within the amorphous regions of spherulites (Figure 13c,d).

On the contrary, the sample prepared from the polymer-rich mixture with 67 wt% of PVDF contains only small submicron pores (Figure 14). As discussed in the previous sections, in such systems, situated to the right of point B, camphor crystallizes only in the presence of a semicrystalline polymer matrix. This takes place in domain III of the phase diagram in Figure 2 in the course of microphase separation within the amorphous polymer regions.

## 4. Conclusions

In this study, we constructed a quasi-equilibrium phase diagram of the PVDF–camphor binary mixture using an original optical method. The diagram includes a boundary curve that describes the dependence of camphor solubility in the amorphous regions of PVDF on temperature. It is argued that the states below the polymer liquidus curve are non-equilibrium, and therefore, it makes little sense to consider phase equilibria in this domain. We also performed a DSC study of the system and plotted the dynamic phase diagram. The diagrams were used to interpret the results of hot-stage microscopy experiments on mixtures cooling from the homogeneous state and SEM experiments on the morphology of capillary-porous bodies obtained upon camphor extraction. The previous assessment of PVDF–camphor systems by Dasgupta et al. [33,34,35] has been found to be inconsistent in different aspects. In particular, both the DSC and XRD results of that study can be explained without assuming the formation of crystalline complexes between the mixture components but by taking into account the size dispersity of polymer and camphor crystals and the high-temperature polymorphic solid–solid transition of camphor. The obtained information can be used as a tool for process engineers dealing with systems where both polymeric and low-molar-mass components reveal crystallinity.

## Figures and Tables

**Figure 1 polymers-14-05214-f001:**
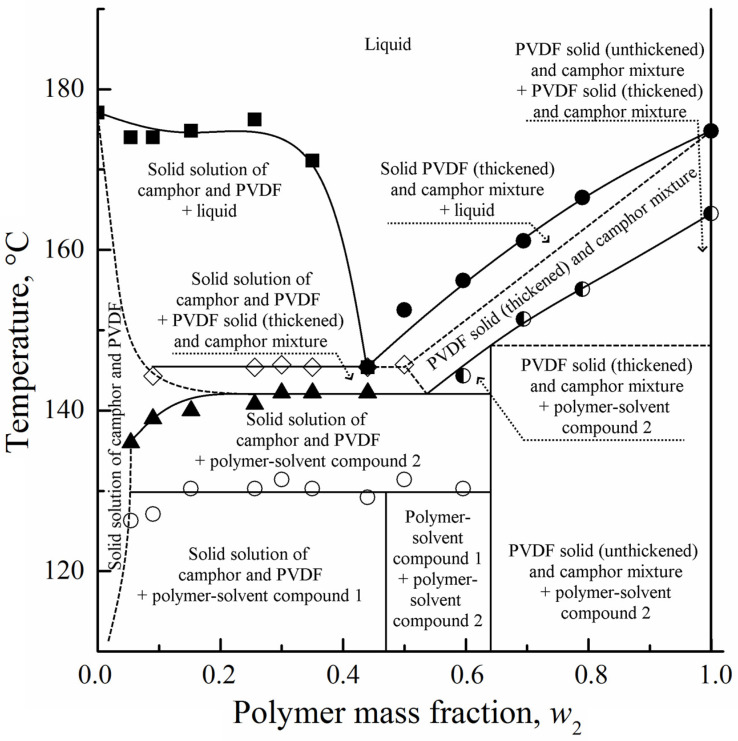
Phase diagram of PVDF–camphor, adapted from [34].

**Figure 2 polymers-14-05214-f002:**
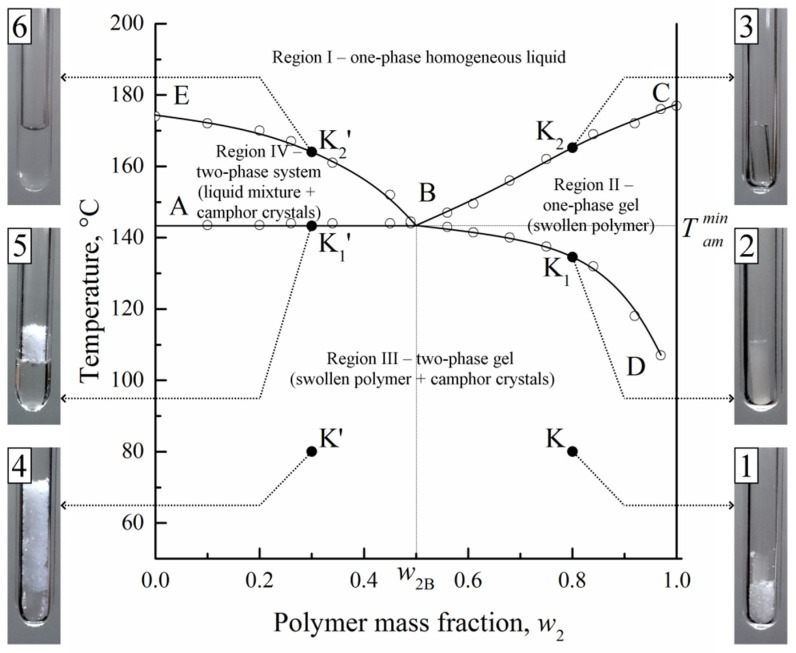
Phase diagram of PVDF–camphor constructed by the optical method and photographs 1–6 illustrating the system state at points K, K_1_, K_2_, K’, K_1_’, and K_2_’.

**Figure 3 polymers-14-05214-f003:**
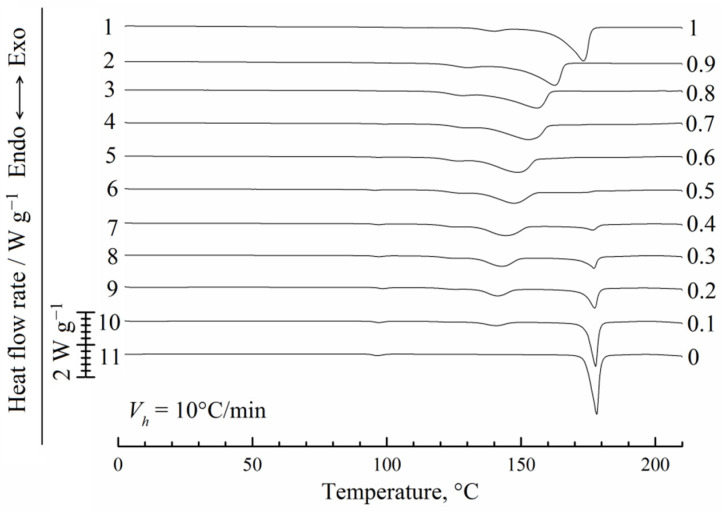
DSC curves of 1st heating for PVDF, camphor, and their mixtures of different compositions. The heating rate is *V_h_* = 10 °C/min. Mass fractions of the polymer are shown to the right of the curves, which are arbitrarily shifted in the vertical direction.

**Figure 4 polymers-14-05214-f004:**
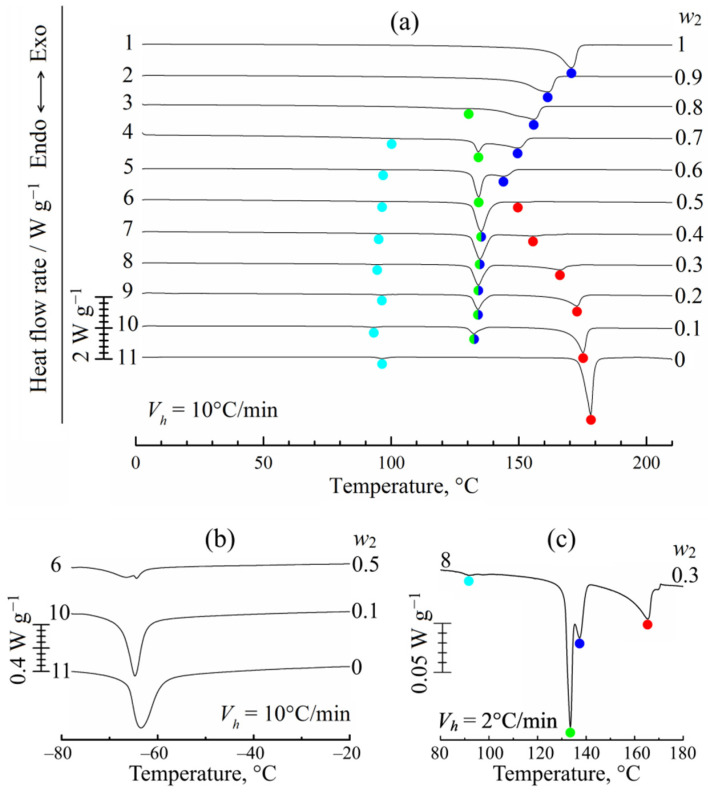
(**a**) DSC curves of the 2nd heating for PVDF, camphor, and their mixtures of different compositions recorded at a heating rate *V_h_* = 10 °C/min. Mass fractions of the polymer, *w*_2_, are shown to the right of the curves, which are arbitrarily shifted in the vertical direction. Color points designate the maximums of endothermic peaks. (**b**) Additional low-temperature endotherm for camphor-enriched mixtures that reflects the solid–solid (orth→hex) transition in camphor. (**c**) One of the DSC curves (*w*_2_ = 0.3) recorded at a heating rate *V_h_* = 2 °C/min.

**Figure 5 polymers-14-05214-f005:**
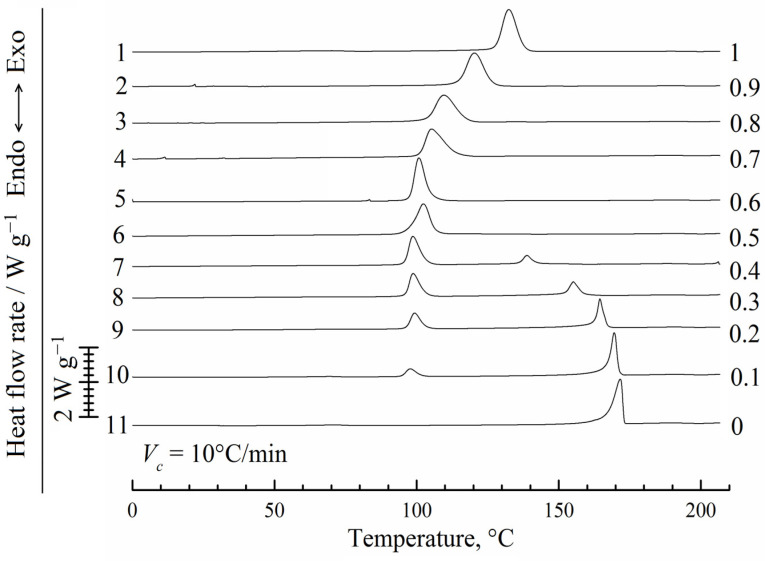
DSC cooling curves for PVDF, camphor, and their mixtures of different compositions recorded at a cooling rate *V_h_* = 10 °C/min. Mass fractions of the polymer are shown to the right of the curves, which are arbitrarily shifted in the vertical direction.

**Figure 6 polymers-14-05214-f006:**
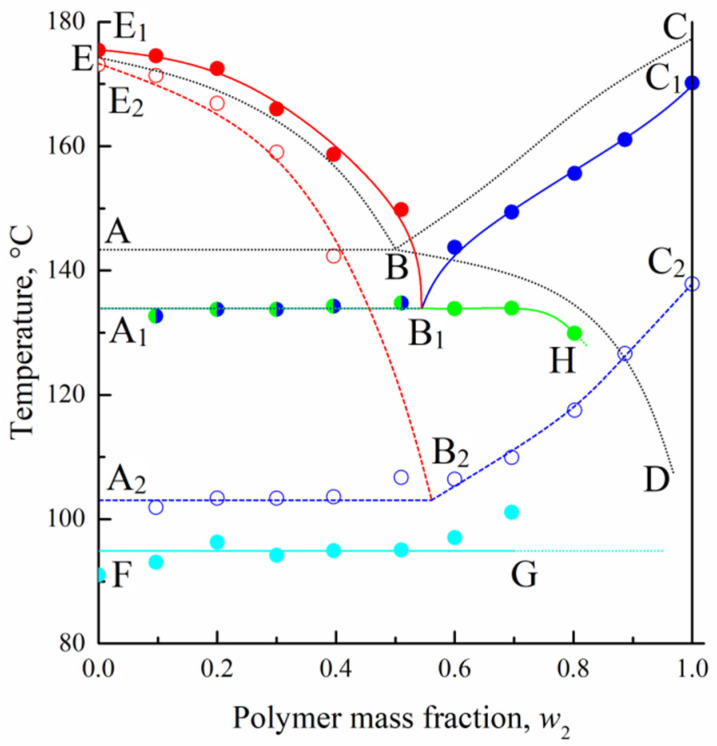
Dynamic phase diagram for the PVDF–camphor system. Solid and open points correspond to DSC 2nd heating and cooling data, respectively. Black dotted curves are replotted from the phase diagram constructed by the quasi-equilibrium optical method (Figure 2). See the text for explanations.

**Figure 7 polymers-14-05214-f007:**
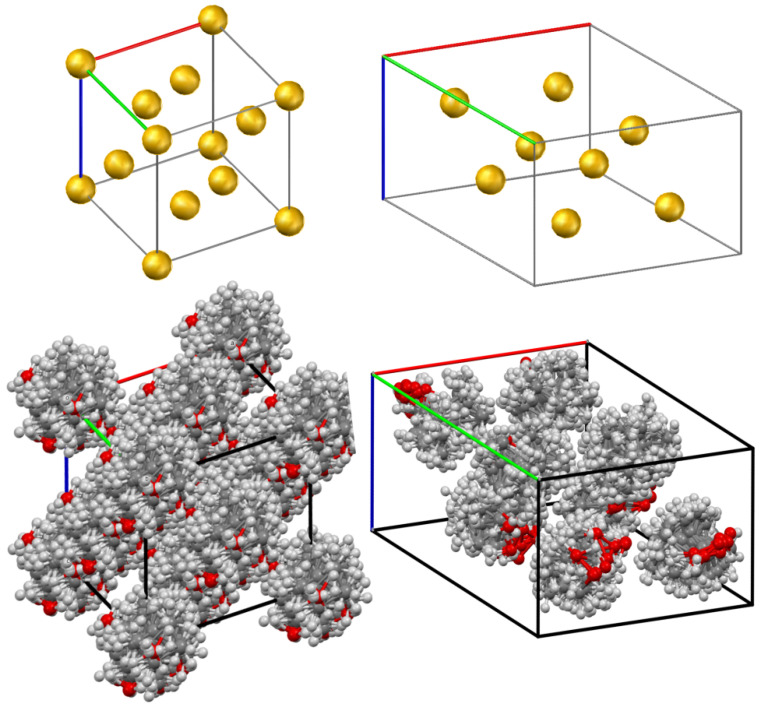
Crystalline structures of the (**a**) high-temperature fcc phase and (**b**) room-temperature hex orientationally disordered phases of camphor. Yellow spheres in the upper figures mark the center-of-mass positions of camphor molecules in the crystalline cells of the fcc and hex phases.

**Figure 8 polymers-14-05214-f008:**
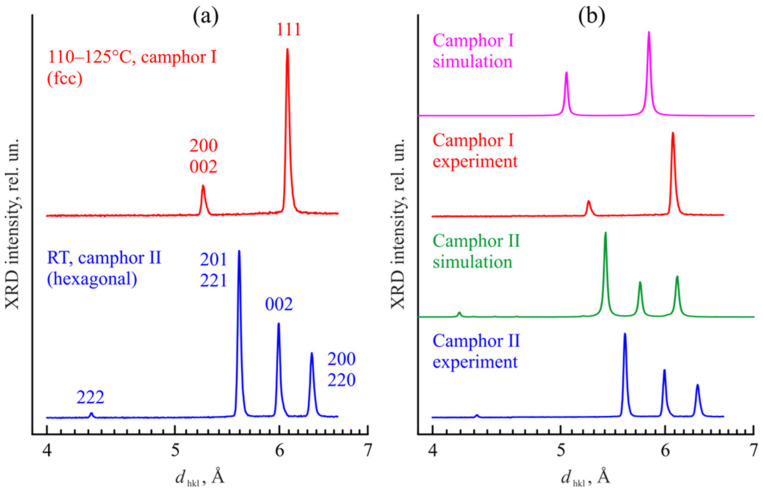
XRD patterns of camphor at high temperature and room temperature: (**a**) experimental data with peak identification and (**b**) comparison with the simulation results based on the fcc (camphor I) and hex (camphor II) phase structures shown in Figure 7.

**Figure 9 polymers-14-05214-f009:**
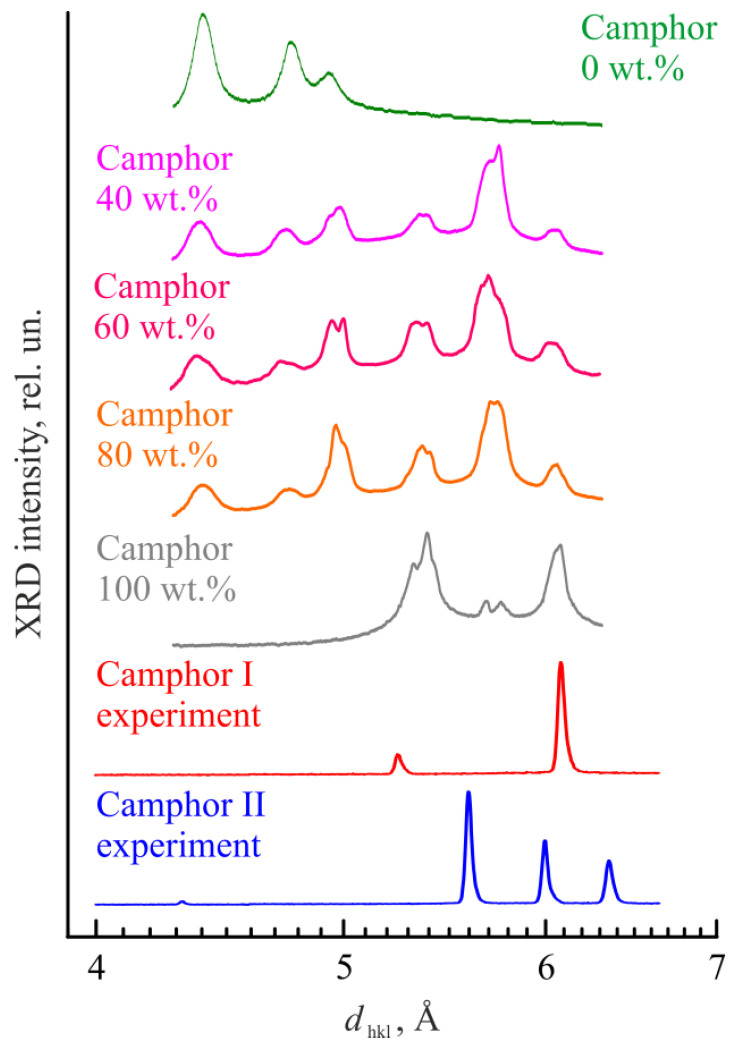
XRD patterns of camphor in the mixtures with PVDF of different compositions at room temperature (adapted from ref. [34]) and XRD patterns of pure camphor at 120 °C (phase I, fcc) and at room temperature (phase II, hex) obtained in this study.

**Figure 10 polymers-14-05214-f010:**
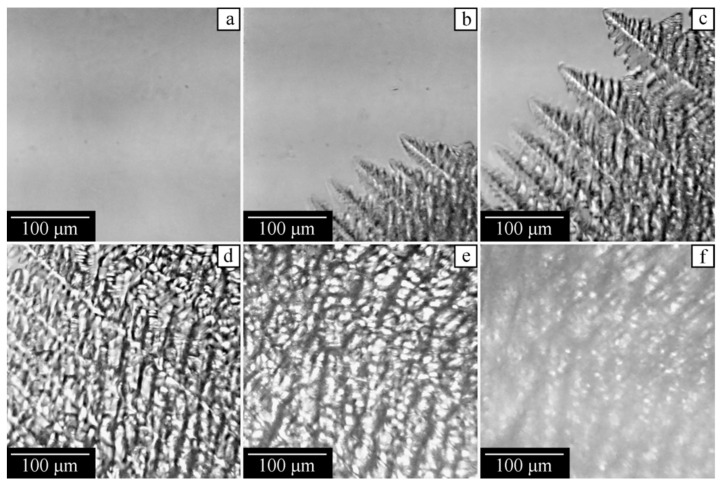
Optical micrographs of the PVDF–camphor mixture with 35 wt% of PVDF taken on cooling from the homogeneous state at (**a**) 180, (**b**) 160, (**c**) 155, (**d**) 150, (**e**) 108, and (**f**) 75 °C.

**Figure 11 polymers-14-05214-f011:**
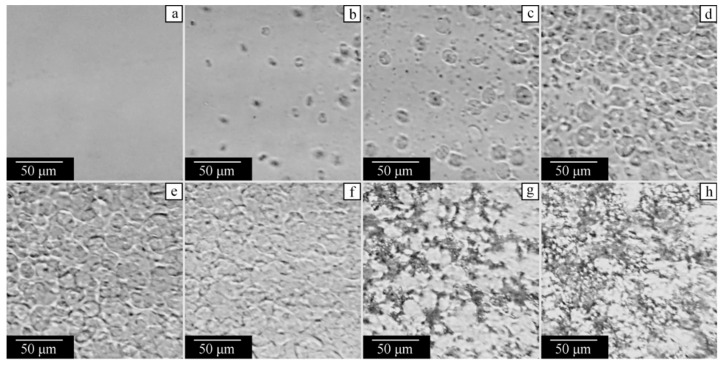
Optical micrographs of the PVDF–camphor mixture with 75 wt% of PVDF taken on cooling from the homogeneous state at (**a**) 180, (**b**) 124, (**c**) 122, (**d**) 119, (**e**) 117, (**f**) 114, (**g**) 106, and (**h**) 75 °C.

**Figure 12 polymers-14-05214-f012:**
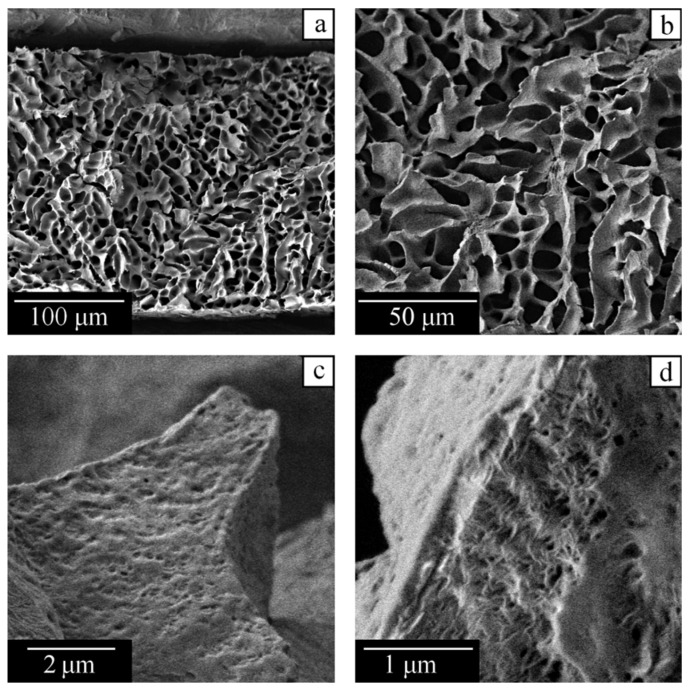
SEM images of the capillary-porous body obtained from the PVDF–camphor mixture containing 28 wt% of the polymer. Note the different scale bars in the images (**a**–**d**).

**Figure 13 polymers-14-05214-f013:**
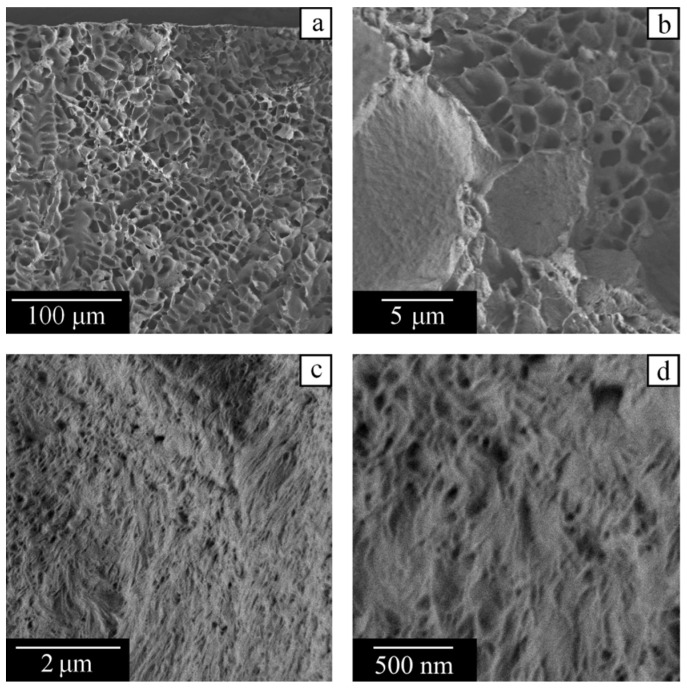
SEM images of the capillary-porous body obtained from the PVDF–camphor mixture containing 45 wt% of the polymer. Note the different scale bars in the images (**a**–**d**).

**Figure 14 polymers-14-05214-f014:**
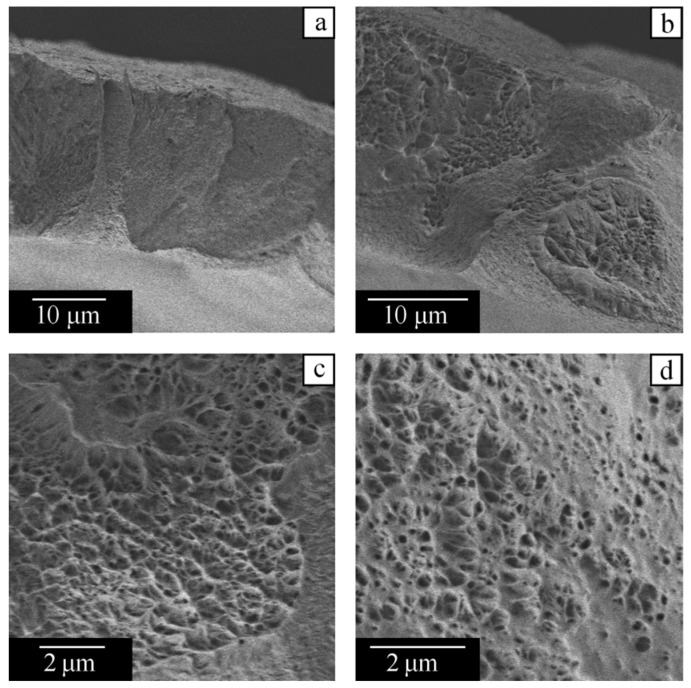
SEM images of the capillary-porous body obtained from the PVDF–camphor mixture containing 67 wt% of the polymer. Note the scale bars in the images (**a**–**d**).

## Data Availability

The data presented in this study are available on request from the corresponding author.

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
