# Peer review of "A New Look at the Structure and Thermal Behavior of Polyvinylidene Fluoride–Camphor Mixtures"

_polymers, 2022, doi:10.3390/polym14235214_

Round 1
Reviewer 1 Report
The experimental quasi equilibrium phase diagram of polyvinylidene fluoride (PVDF) - camphor mixture was constructed by using the original optical method. The phase diagram is used to explain the polarized light hot stage microscope data on cooling the above mixture from a uniform state to room temperature and the scanning electron microscope data on the morphology of the capillary porous body formed after removing camphor. Based on the author's calorimetry and X-ray study, the author described his own views on the possibility of forming inconsistent crystal complexes between PVDF and camphor previously proposed.The work is novel and needs to be modified before the manuscript is published.
1,The two small figures in Figure 4 are very unclear and need to be clear.
2,The author's methods on the evolution of different phases need to refer to higher characterization. For example,Angew. Chem. Int. Ed. 2019, 59, 11642. J. Phys. Chem. Lett. 2021, 12, 12129. Chem. Eng. J. 2020, 125367.
3,What is the basis for the author to use the quasi optical method to construct the equilibrium phase diagram?
4,How did the author get the crystal form in Figure 7? What is the source?
Author Response
Referee 1:
The experimental quasi equilibrium phase diagram of polyvinylidene fluoride (PVDF) - camphor mixture was constructed by using the original optical method. The phase diagram is used to explain the polarized light hot stage microscope data on cooling the above mixture from a uniform state to room temperature and the scanning electron microscope data on the morphology of the capillary porous body formed after removing camphor. Based on the author's calorimetry and X-ray study, the author described his own views on the possibility of forming inconsistent crystal complexes between PVDF and camphor previously proposed. The work is novel and needs to be modified before the manuscript is published.
- The two small figures in Figure 4 are very unclear and need to be clear
Response: Figure 4 was replaced with a new version, in which there are (a), (b), and (c) parts instead of one figure with two insets. The details of the small pictures are now better visible. More explanations were included in the figure caption.
- The author's methods on the evolution of different phases need to refer to higher characterization. For example,Angew. Chem. Int. Ed. 2019, 59, 11642. J. Phys. Chem. Lett. 2021, 12, 12129. Chem. Eng. J. 2020, 125367.
Response: These papers are now mentioned in Section 3.1 on page 6 (ref. 71, 72, and 73).
- What is the basis for the author to use the quasi optical method to construct the equilibrium phase diagram?
Response: Use of the optical method enables one to plot the solubility curve, which is needed to correctly interpret the states of the binary mixtures containing a semicrystalline polymer and a low-molar-mass substance. This was checked by us for several chemically different systems of that kind (ref. [37-44]) and discussed in detail in ref. 38 and in brief in the introduction to this paper. The DSC method can produce dynamic phase diagrams which are closer to or farther from the equilibrium ones depending on the heating rate, but it does not give the solubility curve.
- How did the author get the crystal form in Figure 7? What is the source?
Response: Modeling of the XRD patterns for camphor is described in the last paragraph of the Materials and Methods section on page 5. For the room-temperature hex phase we used the crystallographic information from ref. [50]. For the high-temperature fcc phase we prepared original CIF-files as described in the text. Twenty independent CIF-files are now presented in Supplementary Materials. The text discussing Figure 7 in the first paragraph of Section 3.4 is also expanded.
The English language was checked by a native speaker who corrected several stylistic inaccuracies, but, in fact, he has not found that our original text needed extensive editing. Two other Referees have come to the same conclusion. Nevertheless, we thank Referee 1 for the opportunity to polish the text of our paper.
Reviewer 2 Report
In this contribution, the authors studied the thermal behavior and microstructures of polyvinylidene fluoride (PVDF)-camphor mixtures using microscopy, calorimetry, and X-ray methods. The resulting quasi-equilibrium phase diagram puts doubts in a previous publication. This work encourages the debate and improvement of the PVDF-camphor phase diagram, and this study is inspiring to the readership of Polymers. I would recommend its publishing if the following questions and comments are addressed.
1. What is the molecular weight of PVDF? Is it comparable to the weight-average molecular weight, 180 kg/mol, in Macromolecules 2005, 38, 5602? Does the molecular weight of PVDF impact the compatibility/solubility of PVDF in camphor?
2. What is the effect of heating/cooling rate? For example, the DSC analysis is majorly performed at a heating rate of 10 °C/min, in contrast to 2°C/min in Macromolecules 2005, 38, 5602. Does the different heating/cooling rate result in different crystal sizes of camphor, and different peak positions in DSC, as well as the microstructure in optical graphs, and X-ray patterns?
3. Figure 1 shows a solid solution of camphor and PVDF (Scp in Macromolecules 2005, 38, 5602) at the polymer fraction of 0-0.2 and temperature between 110 and 180 °C. Is this solid solution of camphor and PVDF also observed in this study?
4. In Figure 11, while the number of crystals increases during cooling from 180 to 75 °C, the size of crystals decreases. Is the reduced crystal size because of more nucleation? Does the smaller crystal size create a smaller pore size after removing the camphor? Is the crystal size tunable by the cooling rate?
5. In the study of morphology of the capillary-porous bodies, what is the cooling rate? Do spherulites form during cooling, e.g., near 120 °C as shown in Figure 11c, d, and e? When the PVDF content increases from 28, 45, to 67% in Figure 12, 13, and 14, the mesoporous structure evolves. Did the authors compare the specific surface area, tortuosity, and permeability of the mesoporous PVDF? Figure 14 has the scale bars of 10 and 2 µm, can the 100-µm scale observe any large pores as in Figure 12a?

Author Response
Referee 2:
In this contribution, the authors studied the thermal behavior and microstructures of polyvinylidene fluoride (PVDF)-camphor mixtures using microscopy, calorimetry, and X-ray methods. The resulting quasi-equilibrium phase diagram puts doubts in a previous publication. This work encourages the debate and improvement of the PVDF-camphor phase diagram, and this study is inspiring to the readership of Polymers. I would recommend its publishing if the following questions and comments are addressed.
- What is the molecular weight of PVDF? Is it comparable to the weight-average molecular weight, 180 kg/mol, in Macromolecules 2005, 38, 5602? Does the molecular weight of PVDF impact the compatibility/solubility of PVDF in camphor?
Response: We used commercially available PVDF Solef 6020 from Solvay Specialty Polymers. Three independent studies (ref. [54-56] were added), where the same polymer was used, reported Mw around 650 kg/mol. This information was added to the beginning of the Materials and Methods section on page 3. As generally accepted, the dependence of the boundary lines positions in the polymer-solvent phase diagrams on MM should be negligible for polymers with MM exceeding 100 kg/mol.
- What is the effect of heating/cooling rate? For example, the DSC analysis is majorly performed at a heating rate of 10 °C/min, in contrast to 2°C/min in Macromolecules 2005, 38, 5602. Does the different heating/cooling rate result in different crystal sizes of camphor, and different peak positions in DSC, as well as the microstructure in optical graphs, and X-ray patterns?
Response: Scanning speed in the dynamic experiments on semicrystalline polymers, including DSC, time-resolved SAXS, and hot-stage optical microscopy, does affect peak positions. Therefore, we consider the slowest optical method used by us to be the most accurate. The phase diagram constructed with it helped us to interpret the DSC data consistently, despite the fact that they were indeed obtained at a higher scanning rate than in the aforementioned article in Macromolecules. We give an example of the influence of the heating rate on the position of the peaks in the DSC thermogram in Figure 4. This figure shows that a decrease in the scanning rate to 2 °C/min leads to splitting of the peak located in the region of 133 °C (curve 8 in Figure 4a) into two peaks (Figure 4c). The first, low-temperature peak, which retained its position, reflects the thermal effect of melting the camphor crystals appeared upon cooling the pre-homogenized mixture, and the second, which shifted to higher temperatures, reflects the thermal effect of polymer amorphization in the presence of camphor dissolved in it. Taking into account that the temperature of the maximum of the first peak remained practically unchanged and equal to ~133°Ð¡, it can be concluded that a fivefold decrease in the cooling rate of a homogeneous mixture does not lead to a change in the size of small camphor crystals that arise under such conditions. The size of those crystals is ~10nm in accordance with the Gibbs-Thomson equation.
- Figure 1 shows a solid solution of camphor and PVDF (Scp in Macromolecules 2005, 38, 5602) at the polymer fraction of 0-0.2 and temperature between 110 and 180 °C. Is this solid solution of camphor and PVDF also observed in this study?
Response: We did not find any evidence of the formation of a solid solution between camphor and PVDF. Moreover, since the authors of the above paper in Macromolecules did not explain its structure, it is not entirely clear what they had in mind (probably, something similar to the C1 and C2 complexes discussed in the same work, but with a different stoichiometry).
- In Figure 11, while the number of crystals increases during cooling from 180 to 75 °C, the size of crystals decreases. Is the reduced crystal size because of more nucleation? Does the smaller crystal size create a smaller pore size after removing the camphor? Is the crystal size tunable by the cooling rate?
Response: If the reviewer meant that the size of nascent crystals decreases in the course of cooling, this is indeed due to an increase in the nucleation density. However, this is not evident from Figure 11 and, therefore, not discussed in the text of our paper. The crystal size corresponds to the pore size upon camphor removal – this is written in the first paragraph of Section 3.6 on page 15. It means that the crystal size is indeed tunable by the cooling rate: faster cooling leads to smaller crystals. Since we did not control the cooling rate in our study, this point is not discussed in the text.
- In the study of morphology of the capillary-porous bodies, what is the cooling rate? Do spherulites form during cooling, e.g., near 120 °C as shown in Figure 11c, d, and e? When the PVDF content increases from 28, 45, to 67% in Figure 12, 13, and 14, the mesoporous structure evolves. Did the authors compare the specific surface area, tortuosity, and permeability of the mesoporous PVDF? Figure 14 has the scale bars of 10 and 2 µm, can the 100-µm scale observe any large pores as in Figure 12a?
Response: The cooling rate in experiments aimed at obtaining capillary-porous bodies was ~5 °C/min, as in the experiments on a hot stage optical microscope. Accordingly, the temperatures of polymer crystallization were approximately the same as in the optical experiments (Figures 10 and 11). We have previously studied the specific surface area of such materials by low-temperature nitrogen adsorption (10.1021/acs.jpcb.9b07475 and 10.1016/j.tca.2019.178499) and came to the conclusion that this method cannot reliably estimate the number of mesopores that appear upon removal of the small low-molar-mass crystals. Therefore, we did not use this method in the current study.
In Figure 14, there are no photographs with a lower magnification due to the fact that the thickness of this sample was slightly more than 20 μm. As follows both from the phase diagram (Figures 2 and 6) and from the optical data (Figure 11), the formation of large pores in this sample is impossible, since in the composition region to the right of point B, the polymer always crystallizes first with the formation of a physical gel and, as it cools further, only small crystals of camphor are formed out of this gel.
Reviewer 3 Report
The manuscript entitled: “A new look at the structure and thermal behavior of polyvinylidene fluoride – camphor mixtures” is relevant to the Polymers journal. The article is based on original experimental research. The manuscript was written appropriately for a research article the abstract is sufficiently informative. It should be emphasized that the presented results were described very precisely, and the drawings were prepared understandably. Overall, the paper is well-prepared and should be published as it is. Nevertheless, the article required small improvements:
Fragment 118-124 - Are all the parameters presented in the paper [54]?
Line 134 – Please provide the model and manufacturer of the apparatus.
Line 149 – Is RNMK-05 an optical microscope? You can write it like this, not an observation device.
Line 205 – Perhaps in the description below the Figure, it would be good to mention that it is about photos 1-6.
Figure 3 – Maybe I didn’t notice it, but what is Vh? This was not mentioned in the text. There is also no Y axis, so according to the given scale, it can be assumed that between 1-3, 2-4,3-5 etc., there is 2 W g-1. Additionally, maybe curves 1-11 should describe in the text - how much camphor content they have.
Figure 4 – Maybe instead of inserting insets, it is better to make a drawing with division into a, b, c.
Fragment 210-223 – All photos except number 4 are described. Is this on purpose? What’s in this photo?
Figure12 – Please enter in the description what is in figures a, b, c, d.
Figure 13 i Figure 14 - Please enter in the description what is in figures a, b, c, d.
Author Response
Referee 3:
The manuscript entitled: “A new look at the structure and thermal behavior of polyvinylidene fluoride – camphor mixtures” is relevant to the Polymers journal. The article is based on original experimental research. The manuscript was written appropriately for a research article the abstract is sufficiently informative. It should be emphasized that the presented results were described very precisely, and the drawings were prepared understandably. Overall, the paper is well-prepared and should be published as it is. Nevertheless, the article required small improvements:
Fragment 118-124 - Are all the parameters presented in the paper [54]?
Response: No, it concerns only the melting enthalpy of PVDF at 100% crystallinity – now corrected
Line 134 – Please provide the model and manufacturer of the apparatus.
Response: Provided
Line 149 – Is RNMK-05 an optical microscope? You can write it like this, not an observation device.
Response: Changed to “horizontal microscope”
Line 205 – Perhaps in the description below the Figure, it would be good to mention that it is about photos 1-6.
Response: Description of Figure 2 and its caption were corrected
Figure 3 – Maybe I didn’t notice it, but what is Vh? This was not mentioned in the text. There is also no Y axis, so according to the given scale, it can be assumed that between 1-3, 2-4,3-5 etc., there is 2 W g-1. Additionally, maybe curves 1-11 should describe in the text - how much camphor content they have.
Response: Vh is the heating rate and the DSC curves are arbitrarily positioned in the vertical direction – now mentioned in the captions to Figures 3-5. The polymer weight fraction, w2, is written directly in the Figure next to each curve. Thus, the camphor content is 1–w2. The camphor content is added to the description of Figure 3.
Figure 4 – Maybe instead of inserting insets, it is better to make a drawing with division into a, b, c.
Response: Vh is the heating rate and the DSC curves are arbitrarily positioned in the vertical direction – now mentioned in the caption to Figure 3. The polymer weight fraction, w2, is written directly in the Figure next to each curve. Thus, the camphor content is 1–w2. The camphor content is added to the description of Figure 3.
Fragment 210-223 – All photos except number 4 are described. Is this on purpose? What’s in this photo?
Response: No, this is by mistake – now photo 4 is mentioned in the text.
Figure12 – Please enter in the description what is in figures a, b, c, d.
Response: There are different scale bars in the images – now added to the caption to Figure 12.
Figure 13 i Figure 14 - Please enter in the description what is in figures a, b, c, d.
Response: There are different scale bars in the images – now added to the captions to Figures 13 and 14.
Round 2
Reviewer 1 Report
The revised manuscript has been significantly improved and could be publishable in its present form.